# Genomic Insights into *ARR* Genes: Key Role in Cotton Leaf Abscission Formation

**DOI:** 10.3390/ijms26157161

**Published:** 2025-07-24

**Authors:** Hongyan Shi, Zhenyu Wang, Yuzhi Zhang, Gongye Cheng, Peijun Huang, Li Yang, Songjuan Tan, Xiaoyu Cao, Xiaoyu Pei, Yu Liang, Yu Gao, Xiang Ren, Quanjia Chen, Xiongfeng Ma

**Affiliations:** 1Engineering Research Centre of Cotton, Ministry of Education, College of Agriculture, Xinjiang Agricultural University, 311 Nongda East Road, Urumqi 830052, China; 19194158389@163.com; 2State Key Laboratory of Cotton Bio-Breeding and Integrated Utilization, Institute of Cotton Research, Chinese Academy of Agricultural Sciences, Anyang 455000, China; wangzhenyu8689@126.com (Z.W.); 18839789327@163.com (Y.Z.); cgy1069328061@163.com (G.C.); huangpeij12131@163.com (P.H.);; 3Western Agricultural Research Center, Chinese Academy of Agricultural Sciences, Changji 831100, China

**Keywords:** *ARR* genes family, *Gossypium hirsutum*, VIGS, leaf abscission

## Abstract

The cytokinin response regulator (*ARR*) gene is essential for cytokinin signal transduction, which plays a crucial role in plant growth and development. However, the functional mechanism of *ARR* genes in cotton leaf abscission remains incompletely understood. In this study, a total of 86 *ARR* genes were identified within the genome of *Gossypium hirsutum*. These genes were categorized into four distinct groups based on their phylogenetic characteristics, supported by analyses of gene structures and conserved protein motifs. The *GhARR* genes exhibited an uneven distribution across 25 chromosomes, with three pairs of tandem duplication events observed. Both segmental and tandem duplication events significantly contributed to the expansion of the *ARR* gene family. Furthermore, numerous putative cis-elements were identified in the promoter regions, with hormone and stress-related elements being common among all 86 *GhARR*s. Transcriptome expression profiling screening results demonstrated that *GhARR*s may play a mediating role in cotton’s response to TDZ (*thidiazuron*). The functional validation of *GhARR16*, *GhARR43*, and *GhARR85* using virus-induced gene silencing (VIGS) technology demonstrated that the silencing of these genes led to pronounced leaf wilting and chlorosis in plants, accompanied by a substantial decrease in petiole fracture force. Overall, our study represents a comprehensive analysis of the *G. hirsutum ARR* gene family, revealing their potential roles in leaf abscission regulation.

## 1. Introduction

Organ abscission constitutes a natural process integral to plant growth and development, significantly influencing the regulation of crop yield and the plant life cycle [1]. The abscission process is initiated by the formation of an abscission zone (AZ), a specialized tissue characterized by small, densely packed cells that maintain intercellular communication through plasmodesmata [2]. Upon activation by abscission signals, there is a significant upregulation in the activity of cell wall hydrolases, such as polygalacturonase and cellulase, which leads to the degradation of the middle lamella and primary cell wall, ultimately facilitating organ separation [3,4,5]. In plants, the regulation of abscission is predominantly governed by hormonal signaling, with cytokinin acting as a critical negative regulator that significantly delays leaf senescence and inhibits organ abscission [6,7]. Research indicates that cytokinin response factor 6 (CRF6), whose expression is specifically induced by cytokinin, plays a role in delaying leaf senescence through the AHK3 receptor-mediated signaling pathway [8].

Previous research has elucidated that cytokinin signal transduction is integral to plant growth and development, predominantly facilitated by the Two-Component System (TCS) [9]. The histidine kinase receptor AHK, situated on the plasma membrane, binds to cytokinin in its free base form and undergoes autophosphorylation. This phosphoryl group is then transferred to AHP, which translocates into the nucleus to further convey the phosphoryl group to type-B *ARR* proteins. The phosphorylated type-B *ARR*s act as transcription factors, modulating the expression of target genes, including type-A *ARR*s and CRFs (Cytokinin Response Factors) [10,11,12,13,14]. Subsequently, the dephosphorylated AHP returns to the cytoplasm, where it continues to facilitate phosphoryl group transfer, thereby ensuring the efficient continuation of this signaling cascade. Based on the analysis of amino acid sequences and domain characteristics, Arabidopsis Response Regulator (*ARR*) genes are categorized into four distinct types: type-A, type-B, type-C, and pseudo-*ARR*s (*APRR*s) [15,16]. Type-A response regulators serve as primary cytokinin-responsive repressors, facilitating negative feedback regulation within cytokinin signaling pathways [17]. Conversely, type-B *ARR*s function as transcriptional activators, enhancing the expression of cytokinin-responsive genes and playing pivotal roles in the growth and development of roots and shoots [18,19]. The functional roles of type-C *ARR*s are less extensively studied, yet they are also implicated in the modulation of cytokinin signaling [20]. Additionally, pseudo-*ARR*s (*APRR*s) are predominantly associated with plant circadian rhythms and photoperiodic responses [21].

Research has shown that *ARR* genes are integral to cytokinin signal transduction in plants. Specifically, *ARR4* plays a pivotal role in modulating light signal input into the Arabidopsis circadian rhythm through its interaction with phytochrome B (phyB), a function that is redundant with its homolog *ARR3* [22]. *ARR7* and *ARR15* are involved in the maintenance and regeneration of the shoot apical meristem (SAM) by inhibiting cell proliferation [23]. Moreover, *ARR6*, *ARR7*, and *ARR15* collectively function to negatively regulate seed germination [24]. In the context of signal transduction, *ARR16* is implicated in the coordinated regulation of cytokinin, light, and jasmonic acid signaling pathways, facilitating hypocotyl elongation under blue light conditions [25]. Additionally, *ARR16/17* collaborates with CLE9/10 and SPCH to influence epidermal cell-type composition, thus ensuring leaf adaptability in variable environments [26]. Moreover, *ARR*s have been documented to play a significant role in plant stress responses. Specifically, *ARR1/10/12* contribute to enhanced drought resistance through various mechanisms, such as the induction of anthocyanin biosynthesis, increased sensitivity to abscisic acid (ABA), and the reduction in stomatal aperture [27]. In the context of cold stress, *ARR1* functions as a positive regulator by specifically activating the expression of cold-inducible type-A *ARR*s, including *ARR5/6/7/15* [28]. These findings underscore the crucial role of *ARR* genes in the environmental adaptation of plants.

In cotton production, the utilization of chemical defoliants has emerged as a predominant strategy for managing leaf abscission. Pre-harvest application of these chemical agents facilitates cotton leaf abscission and boll opening, thereby significantly reducing the trash content in raw cotton and enhancing harvesting efficiency [29,30]. TDZ, a synthetic compound with cytokinin-like activity, is a principal active ingredient in cotton defoliants and has been extensively adopted across major cotton-producing regions globally [31]. Research has shown that TDZ promotes leaf abscission through various mechanisms, such as increasing the activity of cell wall-degrading enzymes, stimulating ethylene biosynthesis, and modulating the interaction between cytokinin and ethylene signaling pathways [32]. Nonetheless, the molecular mechanisms governing TDZ-induced *ARR* gene expression and its role in cotton leaf abscission require comprehensive elucidation.

In this study, we performed a comprehensive genome-wide identification and characterization of the *ARR* gene family in *G. hirsutum*, examining their phylogenetic classification, chromosomal localization, and gene structure. Through transcriptome data analysis, one candidate *GhARR* genes were identified. Subsequent VIGS experiments confirmed their significant regulatory roles in the response of cotton to TDZ treatment. These findings elucidate the molecular mechanisms underlying cotton leaf abscission and offer valuable genetic resources for the breeding of machine-harvestable cotton varieties.

## 2. Results

### 2.1. Identification and Characterization of ARR Family Members

Utilizing the protein sequences of 33 *Arabidopsis thaliana ARR* genes as queries, we identified a total of 251 *ARR* genes across four cotton genomes through an integrated approach combining HMMER (version 3.3.2) searches and local BLASTP analysis (Appendix A). The genomic analysis revealed distinct distribution patterns: 43 in *G. arboreum*, 53 in *G. raimondii*, 86 in *G. hirsutum*, and 76 in *G. barbadense* (Additional file: Appendix A). The 86 *ARR* proteins identified in *G. hirsutum* were systematically designated as *GhARR1*-*GhARR86* based on their chromosomal locations (Appendix A). Detailed characterization indicated significant structural diversity among these *GhARR* proteins, with amino acid lengths ranging from 114 to 827 residues (mean length = 449 amino acids), molecular weights from 13.16 to 88.43 kDa, and isoelectric points ranging from 4.65 to 8.82. All members exhibited a negative grand average of hydropathy (GRAVY) values, confirming their hydrophilic nature. Subcellular localization predictions indicated predominant nuclear targeting for most family members, whereas *GhARR2* demonstrated unique cytoplasmic localization, suggesting potential functional specialization.

### 2.2. Phylogenetic Analysis of the ARR Gene Family

To understand the phylogenetic relationship between the *ARR* genes of the four cotton species with *Arabidopsis thaliana*, we constructed a phylogenetic tree containing 291 protein sequences from *G. arboreum* (43), *G. raimondii* (53), *G. barbadense* (76), and *G. hirsutum* (86) (Figure 1). The 291 *ARR* proteins were systematically divided into four groups (I, II, III, and IV), following the classification system used in *Arabidopsis*. Previous research has documented that *Arabidopsis* encompasses 10 type-A *ARR*s, 12 type-B *ARR*s, 2 type-C *ARR*s, and 9 pseudo *ARR*s [15]. Our analysis indicates that Group I corresponds to the type-A *ARR* family, comprising 11 members in *G. arboreum*, 11 in *G. raimondii*, 20 in *G. hirsutum*, and 19 in *G. barbadense*. Group II, identified as the type-B *ARR* family, represents the largest cohort, with 18 members in *G. arboreum*, 23 in *G. raimondii*, 40 in *G. hirsutum*, and 37 in *G. barbadense*. Group III, associated with type-C *ARR*s, is the smallest group, consisting of only 19 members (5 in *G. arboreum*, 4 in *G. raimondii*, 4 in *G. hirsutum*, 4 in *G. barbadense*, and 2 in *A. thaliana*). Group IV comprises pseudo *ARR*s (*APRR*s), with a total of 71 members. It is noteworthy that tetraploid cotton species (*G. barbadense* and *G. hirsutum*) exhibit a significantly greater number of *ARR* genes in each subgroup compared to their diploid counterparts (*G. arboreum* and *G. raimondii*), a pattern likely attributable to genome expansion resulting from polyploidization events.

### 2.3. Chromosomal Distribution and Synteny Analysis of ARR Genes

Based on genomic annotation data, the 86 *GhARR* genes were mapped to 25 chromosomes of *G. hirsutum*, encompassing the 12 chromosomes of the A sub-genome and the 13 chromosomes of the D sub-genome (Figure 2). The distribution of *GhARR* genes across these chromosomes was uneven. Except for a single *GhARR* gene on chromosomes A04, A06, and D06, a minimum of two *GhARR* genes were present on the remaining chromosomes. Notably, there was a high concentration of genes on chromosomes A05, A10, D05, and D12, with the majority of *GhARR* genes organized in clusters. An investigation into gene duplication events was conducted to gain insights into the expansion of the *GhARR* family. Six *GhARR* genes (*GhARR1*/*GhARR2*, *GhARR30*/*GhARR31*, and *GhARR45*/*GhARR46*) were identified as tandem repeat gene pairs located on chromosomes A01, A10, and D01. Besides tandem duplication, fragment duplication events within the *ARR* gene family were also identified (Figure 2). According to the results, 44 pairs of segmental duplication genes and 15 gene clusters containing 52 *GhARR* genes were identified. These findings suggest that the *GhARR* family expanded in *G. hirsutum*, likely due to gene duplication, with segmental duplication events being a key factor in the evolution of *GhARR*s. To elucidate the evolutionary constraints on the *GhARR* family, the Ka/Ks ratios of the *GhARR* gene pairs were determined. For all segmental and tandem duplicated *GhARR* gene pairs, a Ka/Ks ratio of less than 1 was observed (Appendix A), indicating that the *ARR* gene family in *G. hirsutum* has undergone significant purifying selection during its evolutionary history.

The phylogenetic mechanisms of the *G. hirsutum ARR* family were further explored by constructing comparative syntenic maps of *G. hirsutum* associated with *G. barbadense*, *G. raimondii*, and *G. arboreum* (Figure 3). On the whole, 31 *GhARR*s showed a syntenic relationship with those in *G. barbadense*, followed by *G. raimondii* (44) and *G. arboreum* (36). The number of orthologous gene pairs between *G. hirsutum* and the other species (*G. barbadense*, *G. raimondii*, and *G. arboreum*) was 259, 147, and 118, respectively (Appendix A). To characterize the selective pressures acting on these homologous gene pairs during evolution, we calculated the ratio of nonsynonymous to synonymous substitution rates (Ka/Ks) (Appendix A). Remarkably, all homologous pairs exhibited Ka/Ks ratios < 1, with the exception of one gene pair between *G. hirsutum* and *G. barbadense*. This evolutionary pattern strongly suggests that the *ARR* gene family has undergone intense purifying selection throughout its evolutionary history, which likely contributes to the maintenance of functional conservation across divergent *Gossypium* species.

### 2.4. Gene Structure and Conserved Motif Analysis

A comprehensive study of the *GhARR* family’s structural evolution was conducted, including phylogenetic analysis, functional motif conservation, exon–intron patterns, and gene structure characterization. Initially, a maximum likelihood phylogenetic tree was constructed using 86 *GhARR* proteins (Figure 4a), categorizing them into four evolutionary groups (Groups I–IV) comprising 20, 40, 4, and 22 members, respectively. Notably, Group II emerges as the largest and most prevalent group, corroborating findings from previous studies [15]. Through conserved domain analysis (Figure 4b), it was observed that all groups (I–IV) possess a REC domain located at the N-terminal. Most members of Group II exhibit both D–D–K motif and MYB-like DNA-binding domains conserved domains; however, eight proteins, specifically *GhARR2/3/23/40/46/50/66/82*, lack the MYB-like DNA-binding domain, suggesting a degree of conservation within cotton. Group IV is predominantly characterized by the presence of the distinctive CCT domain.

To conduct a more comprehensive analysis of the similarities and diversities in the motif composition of *GhARR* proteins, we annotated 12 conserved motifs as predicted by MEME. The results, as illustrated in Figure 4c, indicate that *GhARR* members within the same subfamily exhibit similar motif profiles. Motif3, motif2, and motif1 exhibited a high degree of conservation and were present in the majority of *GhARR* members. The members of Group II and Group IV predominantly contained the greatest number of motifs (n = 7). In contrast, four members of Group III (*GhARR22*, *GhARR65*, *GhARR24*, and *GhARR67*) possessed only two or three motifs. Furthermore, our analysis revealed that certain motifs were specific to different groups. For instance, motifs 5 and 4 were unique to Group II, while motifs 6 and 10 were specific to Group IV. This distinct motif composition may contribute to the functional diversity observed among *GhARR*s.

The analysis of intron–exon structures indicated that the number of introns in the *GhARR*s varied from 1 to 16 (Figure 4d). Notably, all four *GhARR* genes classified within Group III possessed a single intron and were devoid of both 5′ and 3′ untranslated regions (UTRs). Remarkably, *GhARR4*, a member of Group IV, demonstrated an unusual architecture with 16 introns. The majority of *GhARR* genes contained between four and seven introns, with 19 genes having four introns, 23 genes having five introns, two genes having six introns, and nine genes having seven introns. Members of Group I generally exhibited between one and five introns, with the exception of *GhARR30*, which contained an atypical 13 introns and also lacked both 5′ and 3′ UTRs.

### 2.5. Analysis of Promoter Cis-Acting Elements

To enhance our understanding of transcriptional regulation and the potential roles of *GhARR*s in *G. hirsutum*, we predicted the cis-acting elements within the promoters of *GhARR* genes utilizing the PlantCARE database. The distribution and functional categories of these elements are illustrated in Figure 5. A total of 26 distinct cis-acting elements were identified across the *GhARR* promoter regions, which can be primarily classified into three categories: phytohormone response, stress response, and growth and development (Figure 5). Our findings reveal the presence of several phytohormone-responsive cis-elements, including the abscisic acid (ABA)-responsive element (ABRE), auxin-responsive elements (TGA-element and AuxRR-core), jasmonic acid (JA)-responsive elements (CGTCA-motif and TGACG-motif), gibberellin (GA)-responsive element (GARE-motif), and salicylic acid (SA)-responsive element (TCA-element). Notably, the ABA-responsive element (ABRE) and the JA-responsive elements (CGTCA-motif and TGACG-motif) were found to be enriched in more than half of the *GhARR* genes.

This study further identified diverse stress-related cis-acting elements within the promoters. These elements encompass light-responsive motifs (e.g., AE-box, ATCT-motif, and Sp1), as well as stress-specific elements: LTR for salt stress, MBS for drought response, ARE and GC-motif for anaerobic stress, WUN-motif for wounding stress, and TC-rich repeats for defense and stress responses. Notably, light-responsive components (Box 4, G-box, and motif-GT1) and anaerobic expression elements (ARE) were enriched in over half of the *GhARR* genes (Appendix A). These findings suggest that *ARR* genes may play important roles in diverse stress adaptation processes in plants. Beyond these phytohormone-related elements, several development-associated cis-regulatory elements were detected, including RY-element involved in seed-specific expression, MSA-like elements participating in cell cycle regulation, and CAT-box elements functioning in meristem expression. These comprehensive cis-element profiles suggest that *ARR* genes may integrate multiple hormonal signals and developmental cues to regulate various physiological processes in plants.

### 2.6. Expression Profiles of GhARR Genes Under TDZ Stress

To identify *GhARR* genes involved in TDZ stress response, genes exhibiting at least a two-fold change were selected from the transcriptome expression profile. To further explore the functional differences in *GhARR* genes among various cotton varieties, screenings were performed on defoliant-sensitive varieties (CRI12 and BL34) and a defoliant-insensitive variety (LB24). Thirteen *GhARR* genes (*GhARR2*, *GhARR13*, *GhARR16*, *GhARR18*, *GhARR33*, *GhARR37*, *GhARR43*, *GhARR54*, *GhARR58*, *GhARR60*, *GhARR74*, *GhARR81*, *GhARR85*) were identified, all of which demonstrated significant upregulation at a minimum of four time points (Figure 6). Notably, *GhARR2*, *GhARR13*, *GhARR16*, *GhARR18*, *GhARR33*, *GhARR43*, *GhARR58*, *GhARR74*, and *GhARR85* exhibited consistent differential expression across the cultivars (Appendix A), indicating their potential synergistic role in regulating defoliant response and leaf abscission. Based on RNA-seq data analysis, *GhARR16*, *GhARR43*, and *GhARR85* were selected for subsequent functional studies due to their markedly upregulated expression in response to the defoliant across both sensitive and insensitive varieties.

### 2.7. Functional Analysis of GhARR16, GhARR43 and GhARR85 Gene Silencing in Cotton Under TDZ Stress

To uncover the role of *ARR* genes in responses to defoliant stress in cotton, we used VIGS technology to knock down the expression of *GhARR16*, *GhARR43,* and *GhARR85*. When *pCLCrVA:CLA* plants exhibited the bleached-leaf phenotype (Figure 7a), qRT-PCR indicated that the expression levels of each targeted gene in the plants inoculated with the recombinant vector were significantly reduced compared to the control plants, which proved the effectiveness of gene silencing (Figure 7e–g). At the eight-leaf stage, cotton plants treated with TDZ showed that *pCLCrVA:GhARRs* plants experienced more chlorosis, leaf yellowing, and faster abscission than *pCLCrVA:*00 control plants after 72 h (Figure 7b–d). Quantitative assessments demonstrated a significant reduction in petiole abscission strength in the *pCLCrVA:GhARR16*, *pCLCrVA:GhARR43,* and *pCLCrVA:GhARR85* silenced lines relative to the controls (Figure 7h–j). These results suggest that the silencing of *GhARR16*, *GhARR43,* and *GhARR85* enhances cotton’s sensitivity to the defoliant, indicating that these genes are crucial in modulating TDZ responsiveness.

Subsequent biochemical analyses revealed that the silenced plants accumulated significantly higher levels of malondialdehyde (MDA) and hydrogen peroxide (Figure 8a–c,g–i), alongside a marked decrease in total antioxidant capacity (T-AOC) (Figure 8d–f), in comparison to the *pCLCrVA:*00 controls. These findings further illustrate that the silencing of *GhARR16*, *GhARR43,* and *GhARR85* exacerbates cotton’s sensitivity to defoliant-induced stress through oxidative stress pathways.

## 3. Discussion

The role of cytokinins in plant growth and development is critical, encompassing cell division, differentiation, senescence, leaf abscission, and reactions to environmental stresses [32,33]. The CK signaling pathway includes the response regulator *ARR* family, which is categorized into four groups: *ARR-A*, *ARR-B*, *ARR-C*, and *APRR* [34,35,36]. Across multiple plant species, these groups have been identified [34,37,38,39,40]. In the *G. hirsutum* family, the identification and characterization of the functions of *ARR*s remain insufficiently explored. In the present study, a total of 86 *GhARR* genes were identified within the *G. hirsutum* genome through bioinformatics analysis. Their characteristics, expression patterns, and putative functions were examined in detail.

### 3.1. Diverse Characterization of ARR in G. hirsutum

The uneven gene family size of *ARR* was found in different plants. The genome-wide identification revealed 86 *GhARR* genes in *G. hirsutum*, markedly exceeding the complement found in *Arabidopsis* [41], *rice* [38], *Glycine soja* [42], and *maize* [43], demonstrating an exceptional expansion of this gene family in cotton evolution. This expansion likely originates from the allopolyploid nature of the *G. hirsutum* genome. As an allotetraploid, *G. hirsutum* contains two distinct subgenomes (A and D), and this polyploidization event may have facilitated both the expansion and functional diversification of gene families [44]. Furthermore, during long-term evolution, *G. hirsutum* may have undergone multiple rounds of gene duplication events, further increasing the copy number of *ARR* family members. Phylogenetic analysis classified these genes into four distinct subgroups (A-*ARR*, B-*ARR*, C-*ARR*, and *APRR*), consistent with the established categorization of *ARR* genes in *Arabidopsis* [41]. The B-type *ARR*s constituted the most abundant subgroup, all containing the conserved D-D-K motif and MYB-like DNA-binding domains, strongly suggesting their functional role as transcriptional activators in cytokinin signal transduction. This finding is consistent with previous studies demonstrating that B-type *ARR* genes in *Arabidopsis* also harbor these conserved domains and play pivotal roles in cytokinin signal transduction [41]. The diversification of exon/intron patterns played a vital role in the evolution process of many gene families. We found that the majority of *GhARR* genes contained four to seven introns, consistent with that in *Rhododendron delavayi* [45], *licorice* [46], and *rice* [38].

### 3.2. Evolution and Expansion of the ARR Gene Family in Cotton

Gene duplication is an important mechanism for generating new genetic material during the evolution of organisms [47]. In *G. hirsutum*, 44 segmental duplication events were found in *GhARR*s, with no distinct subgroup distributions, suggesting no significant evolutionary differences in *ARR* family distribution. Chromosomal distribution and collinearity analyses showed that *GhARR* genes are non-randomly organized, with notable clustering on certain chromosomes. This uneven distribution, along with both tandem and segmental duplication events, indicates that gene duplication has significantly contributed to the expansion of the *GhARR* family. The dN/dS ratio, or ω, is often used in evolutionary studies to assess selective pressures on amino acid changes [48]. The Ka/Ks ratios for all pairs are <1, suggesting purifying selection in the *ARR* gene family of *G. hirsutum*, indicating high conservation of *GhARR*s.

Cis-acting regulatory elements in gene promoters are crucial for transcriptional regulation and gene expression [49]. This study’s analysis of cis-elements in the *GhARR* gene promoters reveals three main categories: plant hormone-responsive, stress-responsive, and light-responsive/growth-related elements, aligning with previous research in *Rhododendron delavayi* [45], *licorice* [46], and *rice* [38]. Phytohormone-related elements, such as ABRE, CGTCA-motif/TGACG-motif, and TCA-element, are common in *ARR* family members. Many *ARR* genes also contain stress-related cis-elements like MBS, LTR, G-box, Box 4, and ARE, indicating their role in abiotic stress responses. Notably, 78 cis-acting elements are linked to light responses, highlighting the importance of *GhARR* genes in regulating plant responses to light. Overall, the *ARR* gene family likely plays key roles in both development and stress adaptation in upland cotton.

### 3.3. The Potential Roles of ARR Genes in Response to TDZ Stress

Cotton (*Gossypium hirsutum*) is recognized as one of the most economically significant crops globally. Chemical defoliation constitutes a critical agricultural practice in cotton cultivation. The uniform application of chemical agents that induce defoliation at the appropriate time prior to harvest can enhance defoliation efficiency and reduce the trash content in cotton [32,50]. TDZ, a synthetic molecule with cytokinin-like properties, is extensively utilized as a defoliant to facilitate mechanical harvesting across various crops, particularly cotton [29]. In this study, 13 *GhARR* genes were identified through transcriptome sequencing analysis conducted at seven distinct time points, all of which exhibited a marked upregulation trend at a minimum of four time points. Functional validation via virus-induced gene silencing (VIGS) demonstrated that *GhARR16*, *GhARR43*, and *GhARR85*, belonging to the type-A *ARR* subfamily, are crucial in TDZ-induced leaf abscission. Silencing these genes results in accelerated leaf abscission and a notable reduction in petiole fracture strength. Despite the VIGS experiments clearly confirming the phenotypic roles of *GhARR*s, the molecular mechanisms by which they regulate the abscission process within AZ remain to be elucidated. A comprehensive understanding of the molecular mechanisms underlying cotton leaf abscission holds significant theoretical and practical implications for enhancing cotton harvesting efficiency.

Prior research has demonstrated that *ARR*s are integral components of the cytokinin signal transduction pathway. Within this group, type-B *ARR*s, which function as Myb-like transcription factors, serve as positive regulators of de novo organogenesis and shoot apical meristem (SAM) activity. Certain members of this subgroup can activate the transcription of the *WUS* gene, whereas *ARR1* specifically inhibits shoot regeneration. Conversely, type-A *ARR*s perform negative regulatory roles [51]. In studies concerning the regulation of cotton defoliation, the interaction between cytokinin and ethylene signaling pathways has been substantiated. Specifically, the inhibition of *GhCKX3* has been shown to delay defoliation and attenuate ethylene responses [32]. The abscission marker gene *GhRLF1*, which encodes cytokinin oxidase/dehydrogenase, plays a role in the regulation of leaf abscission through the degradation of cytokinins [50]. Furthermore, TDZ preferentially activates ethylene synthesis genes (*ACOs/ACSs*) and signaling genes (*ETRs/EINs/ERFs*) in leaves, facilitating the transmission of ethylene signals to the abscission zone, thereby promoting the shedding of cotton leaves [52]. This study has limitations, having only identified *GhARR* gene responses to defoliants via transcriptome analysis and initial functional validation. Future research will investigate the physiological processes of *TDZ*-induced cotton defoliation, focusing on the specific roles and molecular mechanisms of *GhARR*s to enhance understanding of cytokinin-ethylene signaling in cotton defoliation regulation.

## 4. Materials and Methods

### 4.1. Identification and Sequence Analysis of the ARR Gene Family

Genomic data for the four cotton species were sourced from the COTTON GEN (https://www.cottongen.org/ (accessed on 29 July 2024)) database, specifically *G. barbadense* (version ZJU V1.1_a1), *G. hirsutum* (version ZJU2.1), *G. arboreum* (version HAU_v2), and *G. raimondii* (version HAU_v2). To thoroughly identify *ARR* genes in cotton, we utilized two complementary methodologies. Initially, we conducted a genome-wide screening using HMMER s (version 3.4; https://www.ebi.ac.uk/Tools/hmmer/home (accessed on 29 July 2024)) with default settings against the cotton proteome, employing hidden Markov model (HMM) profiles of *ARR* domains (PF00072 and PF00498) obtained from the Pfam database (https://www.ebi.ac.uk/interpro/search/sequence/ (accessed on 29 July 2024)). To improve the detection of distantly related homologs while preserving functional relevance, we applied relaxed thresholds (E-value ≤ 1 × 10^−10^ as opposed to the default 1 × 10^−5^) with a minimum sequence alignment coverage requirement of ≥50%. Secondly, we performed localized BLASTP searches utilizing 33 experimentally validated Arabidopsis *ARR* protein sequences obtained from TAIR (https://v2.arabidopsis.org/tools/bulk/sequences/index.jsp (accessed on 29 July 2024))as query inputs against cotton genomic databases, employing stringent criteria (E-value ≤ 1 × 10^−50^, sequence similarity ≥40%). The candidate sequences identified through both HMMER (version 3.4) and BLASTP analyses were consolidated and subjected to redundancy elimination. The remaining sequences underwent further filtering, necessitating the confirmation of complete *ARR* domains (PF00072 or PF0049) in at least two out of three authoritative domain databases: SMART, Pfam, and CDD. Subsequently, we characterized the physicochemical properties of the validated candidates by predicting their theoretical isoelectric points (pI) and molecular weights using the ExPASy proteomics server (https://web.expasy.org/compute_pi/ (accessed on 5 August 2024)) [53]. Additionally, we inferred their subcellular localizations using WoLF PSORT (https://wolfpsort.hgc.jp/ (accessed on 5 August 2024)) [54]. This comprehensive pipeline facilitated the identification of high-confidence *ARR* gene candidates, ensuring both sequence homology and structural integrity.

### 4.2. Phylogenetic Analysis

Multiple sequence alignments of the acquired genes were performed using ClustalW and MEGA (MEGA11) to examine the evolutionary relationships among the *ARR* genes [55]. Using four cotton species (*G. arboreum*, *G. barbadense*, *G. hirsutum*, and *G.raimondii*) along with *Arabidopsis thaliana* as reference, we constructed phylogenetic trees employing the maximum likelihood (ML) method. The resulting trees were visualized using the Interactive Tree of Life (iTOL) platform (https://itol.embl.de/ (accessed on 5 August 2024)) [56].

### 4.3. Analysis of Chromosomal Location and Collinearity of ARR Family Genes in G. hirsutum

The identified *GhARR* genes were mapped onto chromosomes based on the genome’s physical location data. Utilizing its default parameters, MCScanX was employed to analyze tandem and segmental duplication events across the entire genome of *G. hirsutum* [57]. We further conducted genome-wide collinearity analyses between *G.hirsutum* and three related species (*G.barbadense*, *G.arboreum*, and *G. raimondii*) using MCScanX with default parameters, with subsequent visualization performed using TBtools-II (v2.322). To illustrate these findings, TBtools was employed for visualization. The selection pressure on genes encoding *ARR* proteins was assessed by calculating the ratio of nonsynonymous (Ka) to synonymous (Ks) substitutions, with the Ka/Ks ratio serving as an indicator of selective pressure. The values for Ka, Ks, and the Ka/Ks ratio were determined using the Simple KaKs Calculator within the TBtools-II (v2.322) software [58].

### 4.4. Conserved Domain, Gene Structure, and Motif Analysis of the ARRs

To conduct a comprehensive analysis of the gene structures of *GhARR*s, the coding sequence (CDS) and GFF3 format files of *G. hirsutum* were obtained from the cotton genome database. The *ARR* protein sequences were subjected to conserved domain analysis using the Pfam database, NCBI’s Conserved Domain Database (CDD), and SMART. The analysis results were visualized with TBtools. The conserved motifs within the *ARR* proteins of *G. hirsutum* were identified using the MEME suite, specifying the number of motifs as 12 while keeping all other parameters at their default settings. An intron–exon map was generated using TBtools-II (v2.322) software [58].

### 4.5. Expression Analysis of GhARRs

The gene expression data pertaining to *GhARR*s under TDZ treatment were sourced from publicly available transcriptome datasets. Specifically, the RNA-seq data utilized in this study have been published and can be accessed in the Journal of Advanced Research (2023, DOI: 10.1016/j.jare.2023.05.007) and the International Journal of Molecular Sciences (DOI: 10.3390/ijms21082738) [59,60]. Transcript abundance was quantified using fragments per kilobase of exon per million mapped reads (FPKM) values, which were derived from RNA-seq reads. To enhance the visualization of expression level differences and facilitate subsequent statistical analyses, the FPKM values were subjected to log2-transformation. Heatmaps illustrating these data were generated using TBtools-II (v2.322).

### 4.6. Plant Materials and Growing Conditions

In this experiment, the widely cultivated Xinjiang cotton variety ZM113 was chosen as the research subject. Mature, full-grained cotton seeds were selected and immersed in sterile water for 24 h to enhance germination. The potting medium was prepared by combining nutrient soil and vermiculite in a 1:1 volumetric ratio, and this mixture was then distributed evenly into pots measuring 15 cm in diameter and 13 cm in height. Three seeds were sown in each pot. Following seedling emergence and full expansion of the cotyledons, thinning was performed to leave one cotton seedling per pot. The experiment was conducted within a greenhouse environment under meticulously controlled conditions: the temperature was sustained at 28 ± 2 °C; the photoperiod was structured as 12 h of light followed by 12 h of darkness; the light intensity varied between 150 and 250 μmol m^−2^s^−1^, and the relative humidity was maintained between 50% and 60%. Throughout the plant growth period, water and nutrients were supplied adequately to fulfill the growth and developmental requirements of cotton. TDZ treatment was initiated when the cotton seedlings, approximately 30 days after sowing, had developed 8 true leaves. At the eight-leaf stage, the defoliant thidiazuron (TDZ, 100 mg/L) was sprayed evenly on all the leaves of the cotton plants [60]. Sampling and force measurements were performed on the third day post-treatment. Sampling was conducted on the first and second inverted leaves from the plant apex, while force measurements encompassed all leaves. For sampling, three biological replicates were utilized, and 20 plants exhibiting uniform growth were selected for force measurement. All samples were immediately flash-frozen in liquid nitrogen upon collection and stored at −80 °C for subsequent physiological analysis. The inclusion of three biological replicates was employed to ensure the reliability and accuracy of the experimental outcomes.

### 4.7. RNA Extraction and qRT-PCR Analysis

Total RNA was extracted from liquid nitrogen quick-frozen leaf samples using TRIzol reagent; RNA concentration and purity (A260/A280 ratio) were detected using a NanoDrop 2000 spectrophotometer (Shanghai Precision Instruments Co., Ltd., Shanghai, China), and RNA was assessed by 1.5% agarose gel electrophoresis for RNA Integrity. An amount of 1 μg of total RNA was reverse transcribed to cDNA using the Quantitect Reverse Transcription Kit. qRT-PCR reactions were performed using the Taq Pro Universal SYBR qPCR Master Mix (Vazyme, Nanjing, China) in the QuantStudio 5 real-time fluorescence quantitative PCR instrument (Thermo Fisher Scientific, Shanghai, China). The reaction system (20 μL) contained 10 μL 2×SYBR Green Master Mix, 0.8 μL upstream and downstream primers (10 μM), 2 μL cDNA template, and 7.2 μL nuclease-free water. The reaction program was as follows: pre-denaturation at 95 °C for 30 s; denaturation at 95 °C for 10 s, annealing/extension at 60 °C for 30 s, and 40 cycles. Three technical replicates were set up for each sample, and GhUBQ7 was used as the internal reference gene. The relative expression of the target genes was calculated by the 2^−ΔΔCt^ method, and a Student’s *t*-test was performed using SPSS 22.0 software to determine the significance of differences between groups (*p* < 0.01).

### 4.8. Virus-Induced Gene Silencing

*GhARR1*6, *GhARR43,* and *GhARR85* specific coding sequences of 300 bp each were designed according to the gene silencing sequence design website (https://crm.vazyme.com/cetool/multifragment.html (accessed on 10 September 2024). Sequences obtained were used for virus-induced gene silencing (VIGS) experiments. Primers used for amplification are detailed in Appendix A. Target fragments were amplified and cloned into the tobacco rattlesnake virus vector (*pCLCrVA*) using the Clon Express^®^ Ultra One Step Cloning Kit (C115, Vazyme, Nanjing, China). The constructed vectors included *pCLCrVA:GhARR16*, *pCLCrVA:GhARR43,* and *pCLCrVA:GhARR85*, while *pCLCrVA:*00 was set as a negative control. The vector Agrobacterium strain was mixed with the *pCLCrVB* containing Agrobacterium strain in a 1:1 ratio, and the mixture was injected into the cotyledons of cotton seedlings using a 1 mL needleless syringe. After injection, the plants were incubated in the dark for 24 h. Subsequently, the plants were transferred to a thermostatic light chamber (28 °C, 16 h of light/8 h of darkness) for incubation until they grew to the 8-leaf stage (approximately 30 days after sowing). Subsequently, the plants were subjected to TDZ treatment for 72 h, and the leaves were collected for biochemical analysis. For biochemical analysis, malondialdehyde (MDA), total antioxidant capacity (T-AOC), and hydrogen peroxide (H_2_O_2_) contents were quantified using assay kits manufactured by Solebo, specifically MDA (BC0025), T-AOC (BC1315), and H_2_O_2_ (BC3593). Three replicates of each experiment were performed to ensure the reliability of the results.

## 5. Conclusions

In this study, we performed a comprehensive genome-wide analysis of the *ARR* gene family in *G. hirsutum*. Utilizing the genomic data of *G. hirsutum*, we identified a total of 86 *GhARR* genes. Following the classification of *ARR* genes in *Arabidopsis*, these 86 *GhARR* genes were categorized into four types: Type-A, Type-B, Type-C, and Pseudo. Our investigation into gene duplication events revealed six instances of tandem duplication among the 86 *GhARR* genes. Notably, 45 of these genes experienced segmental duplication events, which likely contributed significantly to the expansion of this gene family. Transcriptomic analysis, conducted under various treatment conditions, indicated that the expression levels of *GhARR16*, *GhARR43*, and *GhARR85* were markedly elevated in response to TDZ treatment. Silencing the gene expression of *GhARR16*, *GhARR43*, and *GhARR85*, members of the type-A *ARR*s subfamily, led to chlorosis in cotton plants, a reduction in petiole breaking force, and an increased rate of leaf abscission. This study offers fundamental evidence for elucidating the molecular regulatory network governing cotton leaf abscission. In future research, these findings can be employed to thoroughly investigate gene interactions and their response mechanisms under varying environmental conditions, thereby aiding in the genetic enhancement of cotton agronomic traits.

## Figures and Tables

**Figure 1 ijms-26-07161-f001:**
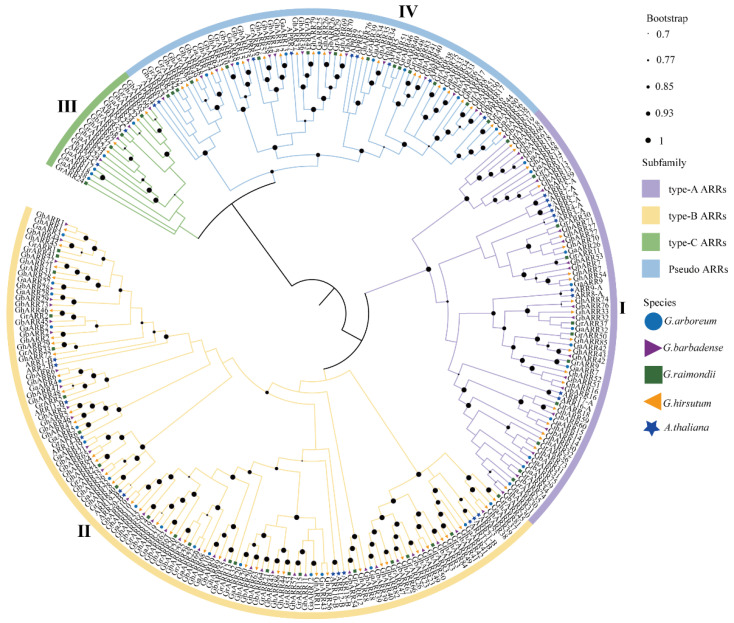
Phylogenetic relationships of 291 *ARR* proteins from *G. hirsutum*, *G. barbadense*, *G. arboreum*, *G. raimondii,* and *A. thaliana*. MEGA11 was used to construct the phylogenetic tree using the maximum likelihood method with 1000 bootstrap replications. Each color indicates an individual group.

**Figure 2 ijms-26-07161-f002:**
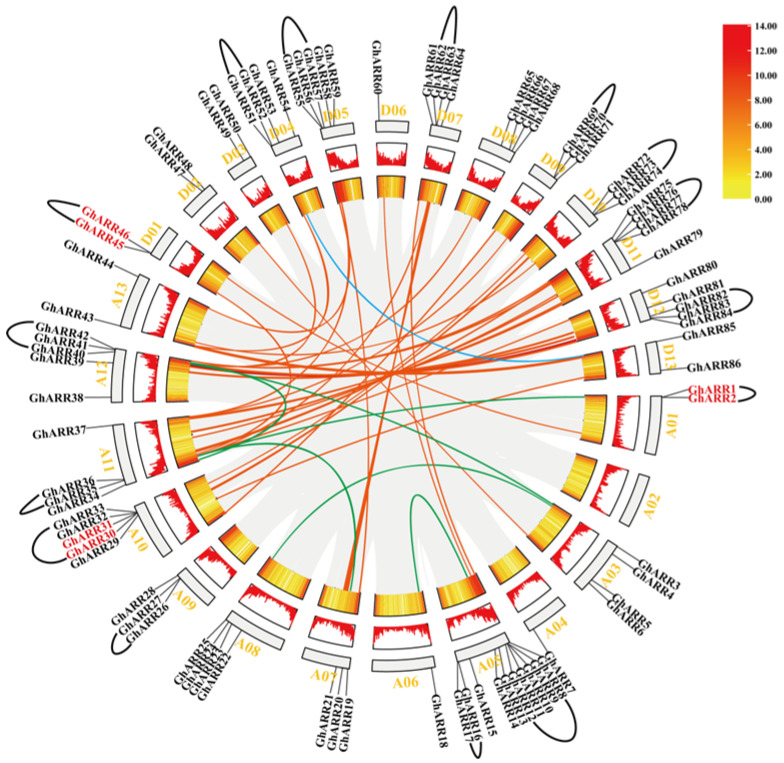
Chromosomal locations and collinear relationships of the *ARR* genes in *G. hirsutum*. The chromosome number is indicated next to each chromosome. Black brackets indicate *ARR* gene clusters. The outermost circular ring shows specific chromosomal localization of *ARR* genes, with red font marking tandemly duplicated genes (tandem duplication). Red lines within the boxes indicate gene abundance distribution in chromosomal regions, where line height correlates positively with abundance values. Orange–yellow boxes represent upland cotton gene density, visualized as a heatmap. The syntenic blocks within the *GhARR* genome are delineated by gray lines. The red, green, and blue lines represent duplicated *ARR* gene pairs between subgenomes: green lines correspond to A–A subgenome pairs, red lines indicate A–D subgenome pairs, and blue lines denote D–D subgenome pairs.

**Figure 3 ijms-26-07161-f003:**
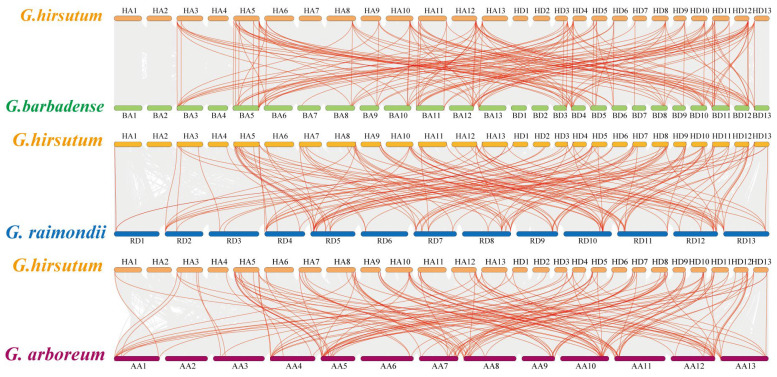
Synteny analysis of *ARR* genes between *G. hirsutum*, *G. barbadense, G. raimondii,* and *G. arboreum*. Gray lines in the background indicated all synteny blocks in the genome, while the red lines indicated the duplication of *ARR* gene pairs.

**Figure 4 ijms-26-07161-f004:**
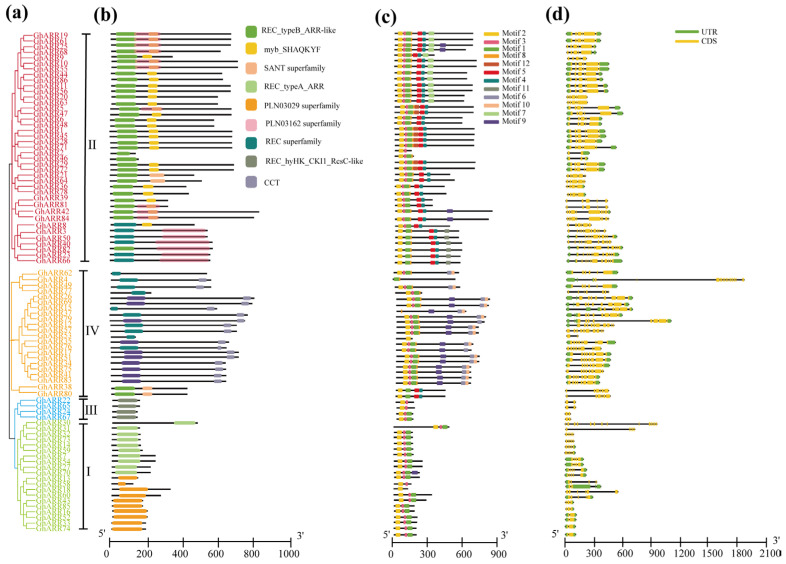
Genetic structure of the *GhARR* gene family in upland cotton. (**a**) Evolutionary Analysis of the *GhARR* Gene Family. (**b**) Analysis of Conserved Domains in the *GhARR* Gene Family. (**c**) Motif Analysis of the *GhARR* Gene Family. (**d**) Structural Analysis of *GhARR* Genes.

**Figure 5 ijms-26-07161-f005:**
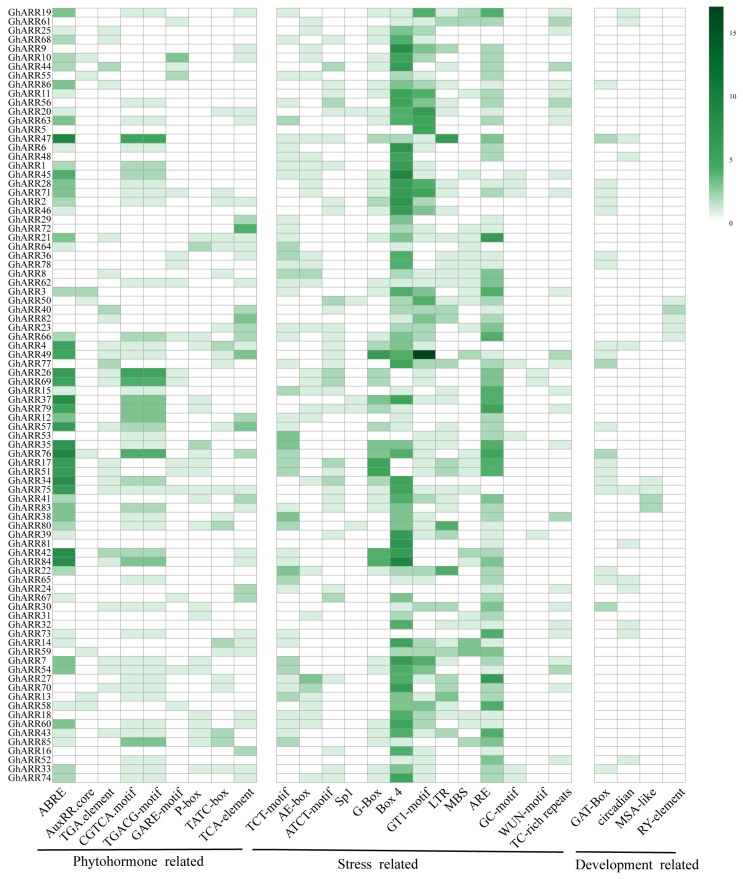
The distribution of cis-acting elements in promoters of *GhARR*s. The corresponding number of them was indicated by the color scale.

**Figure 6 ijms-26-07161-f006:**
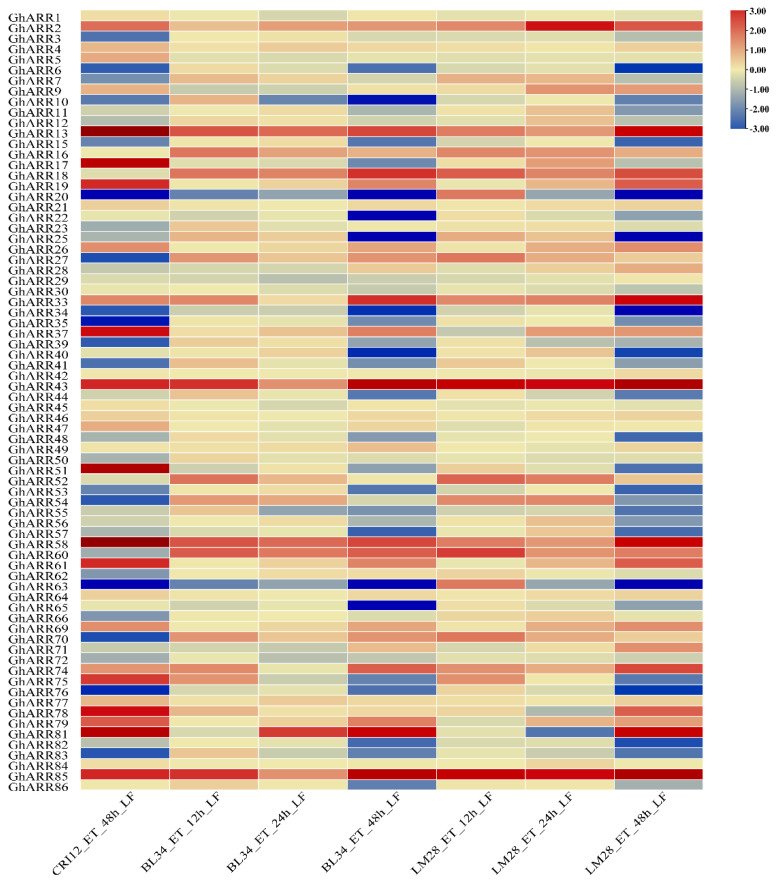
The expression patterns of *GhARR* genes in leaves. The RNA-Seq expression profiles of *G.hirsutum* were used to identify the relative expression levels of *GhARR* genes. The heatmaps were created by TBtools-II (v2.322) based on the transformed data of log2 (FPKM + 1) values, and the cluster analysis was performed on gene expression level by row. Gene expression levels are represented using a color-coded scale. Defoliant-sensitive, CRI12, BL34, and defoliant-insensitive, LM28, leaf parts. The detailed FPKM values are present in Appendix A.

**Figure 7 ijms-26-07161-f007:**
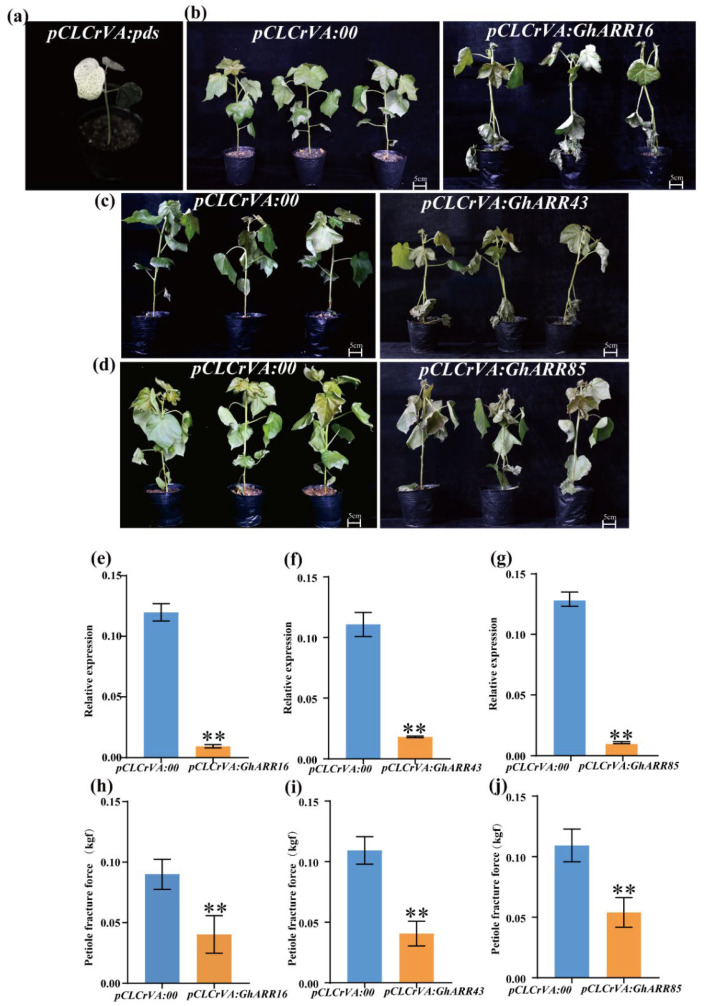
Silencing *GhARR16*, *GhARR43, and GhARR85* enhances sensitivity to defoliants. (**a**) Albino phenotype. (**b**–**d**) Phenotype of plants. Negative control (left), and silenced plant (right). (**e**–**g**) The silencing efficiency of *GhARR16*, *GhARR43, and GhARR85* was detected by qRT-PCR. (**h**–**j**) Petiole fracture force of plants after 72 h of TDZ stress at normal temperature. Student’s *t*-test was employed, and the results were presented as mean ± standard error (SE). Statistical significance was set at the following levels: ** *p* < 0.01, (n = 3).

**Figure 8 ijms-26-07161-f008:**
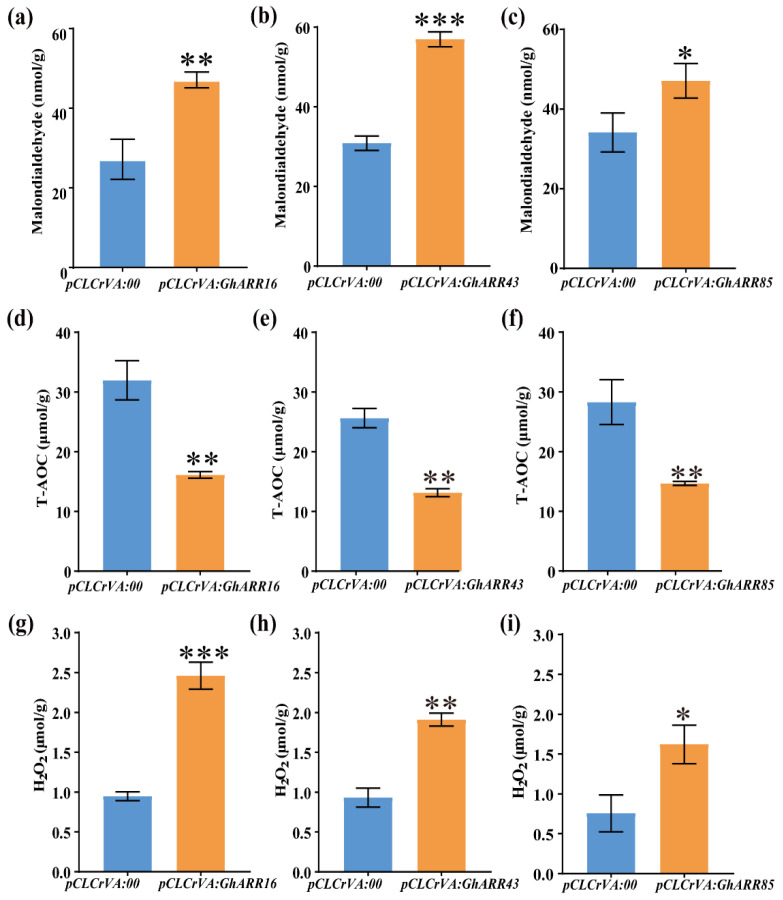
Effects of silencing *GhARR16*, *GhARR43, and GhARR85* on physiological and biochemical parameters of cotton. (**a**–**c**) The content of malondialdehyde in cotton. (**d**–**f**) Total antioxidant capacity (T-AOC) levels in cotton. (**g**–**i**) The content of hydrogen peroxide (H_2_O_2_). Student’s *t*-test was employed, and the results were presented as mean ± standard error (SE). Statistical significance was set at the following levels: * *p* < 0.05, ** *p* < 0.01, and *** *p* < 0.001 (n = 3).

## Data Availability

Data is contained within this article or the Appendix A.

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
