# Peer review of "Genomic Insights into ARR Genes: Key Role in Cotton Leaf Abscission Formation"

_ijms, 2025, doi:10.3390/ijms26157161_

Round 1
Reviewer 1 Report
Comments and Suggestions for Authors
- Abstract: Major results should be presented quantitatively (e.g., specific values) or with explicit identifiers (e.g., gene names), avoiding vague generalizations. For instance, the three genes used in VIGS must be listed. Abbreviations should be defined upon first use, such as thidiazuron (TDZ).
- Introduction: The hyperlink for Reference 1 is missing.
- Results: 2.1. To address concerns about completeness of your methodology employed for identifying homologous genes across cotton genomes, clear description for the designing of the maximize detection sensitivity should be added to minimize the risk of omissions.
2.2. The counts of type-A RR genes in both cotton and Arabidopsis should be explicitly stated. This quantitative comparison will determine whether the distribution pattern of RR subfamilies—where B-type is consistently the most abundant—is conserved in cotton.
2.4. Delete duplicated sentence at Line 215 (original at Line 214)."
2.6. At Line 264, “Transcriptome” not “Tran-scriptome”. Key DEGs should be explicitly listed rather than only reporting the total number of DEGs. What criteria were used to prioritize three candidate genes from thousands of DEGs? Was Table S7 the sole basis for this selection? Were these three genes also differentially expressed in the ZM113-LF or PL RNA-seq datasets?
2.7. How many biological replicates (plants) were used for phenotypic analysis in this experiment?
- Discussion: 3.1. The hyperlink for References in this part are missing.
3.2. What does “ADK Gene Family” mean in the subtitle?
The Discussion primarily rehashes the Results and Introduction. It should be restructured to provide an in-depth interpretation of the findings and propose future research directions.
- Material and Methods: 4.5. If “the RNA-seq data is unpublished”, have they been deposited in a public repository? Were these data generated by you for this study?
4.8. Why the word “pCLCrVA:00” is in Red?
- Conclusion: Since ARR denotes Arabidopsis Response Regulator genes (as defined in your Introduction/Discussion), cotton RR genes should consistently be designated as GhRR. However, ARR and generic RR are used interchangeably throughout the manuscript—especially in the Conclusion—causing confusion. Please standardize terminology to distinguish species-specific gene families.
Reviewer 2 Report
Comments and Suggestions for Authors
This manuscript presents a comprehensive genomic analysis of the ARR gene family in Gossypium hirsutum and functionally validates the role of three GhARRs in thidiazuron (TDZ)-induced leaf abscission. The study is a mix of bioinformatics, transcriptomics, and virus-induced gene silencing (VIGS) to elucidate mechanisms underlying cotton defoliation.
While the manuscript provides substantial genomic insights and functional data, the following concerns need to be addressed by the authors.
- While VIGS confirms phenotypic roles, the molecular pathways linking GhARRs to abscission remain unclear. How do GhARRs modulate cytokinin signaling in AZs? For example, does TDZ alter cytokinin degradation (GhCKX3) or ethylene biosynthesis (GhACS)?
- Are GhARRs direct transcriptional regulators of cell-wall hydrolases (GhCEL1, GhPG)?
- VIGS efficiency was demonstrated via qRT-PCR (mRNA level); however, protein-level validation (e.g., Western blot) would provide stronger evidence for functional gene silencing.
- Figure 6 lacks statistical annotations.
- In Figure 7, error bars for petiole fracture force and qRT-PCR are unclear; specify biological replicates and statistical tests used.
- Specify TDZ concentration.
- There are some minor typos in the manuscript.
- Please check the reported references also in accordance with the format required by “IJMS”
Round 2
Reviewer 1 Report
Comments and Suggestions for Authors
Overall, I am pleased to confirm that the authors have adequately addressed the majority of my previous concerns during the revision. The manuscript has been significantly improved and now presents a much stronger contribution to the field.
Reviewer 2 Report
Comments and Suggestions for Authors
The authors have made the necessary corrections and clarifications to the text of the manuscript. The article may be accepted for publication.